# The Perception of Lactose-Related Symptoms of Patients with Lactose Malabsorption

**DOI:** 10.3390/ijerph191610234

**Published:** 2022-08-17

**Authors:** Michele Di Stefano, Natascia Brondino, Vera Bonaso, Emanuela Miceli, Francesco Lapia, Giacomo Grandi, Elisabetta Pagani, Gino Roberto Corazza, Antonio Di Sabatino

**Affiliations:** 1Department of Internal Medicine, IRCCS “S. Matteo” Hospital Foundation, University of Pavia, 27100 Pavia, Italy; 2Department of Brain and Behavioral Sciences, University of Pavia, 27100 Pavia, Italy

**Keywords:** lactose, lactose intolerance, nocebo effect, self-reported symptom

## Abstract

*Background:* Dairy products are frequently considered responsible for post-prandial symptoms and are withdrawn from the diet, even against medical advice. We analysed the symptoms patients consider as lactose related; we also evaluated if psychological profile may affect the interpretation of the relationship between lactose and symptoms. *Methods:* In 268 patients undergoing lactose breath test, symptoms considered evoked by lactose intake were recorded and their severity measured. In the second part, symptom onset of 40 randomly selected patients was detected after both lactose and glucose breath test were blindly performed. Questionnaires evaluating anxiety, suggestibility and personality trait were administered. *Key Results*: Symptoms depending on functional gastrointestinal disorders or reflux disease were frequent in self-reported lactose-intolerant patients. In comparison with lactose malabsorption, these symptoms proved to be more frequent in patients with negative lactose breath test. The blinded administration of lactose and glucose demonstrated that a correct link between lactose intake and symptom onset was possible, only in 47.5% of the subjects, making this test inaccurate. None of the investigated psychological characteristics were different between patients with a nocebo response and patients not experiencing nocebo. *Conclusions*: Patients with self-reported lactose intolerance are frequently unaware about clinical presentation of this condition, and correct information is needed. The detection of symptom onset after lactose is an inaccurate test for lactose intolerance. Furthermore, the analysis of psychological characteristics of patients undergoing hydrogen breath test is not useful to select the subgroup at risk for a nocebo response. New strategies to diagnose lactose intolerance are mandatory.

## 1. Introduction

Patients with functional gastrointestinal disorders frequently report a strict correlation between food ingestion and gastrointestinal symptom occurrence [1,2]. Milk and other dairy products are considered at the top of the list of foods responsible for the onset of gastrointestinal symptoms [3]; patients frequently withdraw them from the diet, even against medical advice. Food withdrawal from diet is an effective therapeutic measure if an unequivocal causal relationship with symptom or lesion onset is demonstrated, as in coeliac disease, but this is not the case for many patients claiming lactose and other food intolerances [4]. In our experience, there are frequent visits from patients claiming a strict relationship between lactose intake and the occurrence of symptoms currently induced by the presence of other conditions different from lactose intolerance.

A clear responsibility of foods in symptom onset is defined if at least two conditions occur: the specific food may have the ability to induce a specific symptom through a specific pathophysiological mechanism; patients should correctly interpret the association between the intake of this food and the onset of a specific symptom pattern. These two conditions in the clinical practice are not fulfilled in many patients: 30 years ago, in one-third of patients with severe self-reported lactose intolerance, a normal capacity to absorb lactose was shown [5]. Additionally, in patients with lactose malabsorption, after dairy food intake, lactose did not prove to be a major cause of symptoms [6]; finally, in patients undergoing lactose breath test, an important role of nocebo effect in the onset of symptoms was recently demonstrated [7]. These results show that patient’s judgement is frequently incorrect. The hydrogen breath test is accurate to diagnose lactose malabsorption [8,9], but the reporting of symptom onset after oral lactose administration could be inaccurate for diagnosing intolerance, due to the patients’ conviction of ingesting a potentially harmful food, which results in negative expectations about the test [7]. 

The aims of this study were: (a) to evaluate the symptoms patients consider as lactose related; (b) to evaluate the accuracy of symptoms reported during hydrogen breath test to diagnose intolerance; (c) to evaluate if psychological characteristics of the patients may affect the interpretation of the relationship between lactose intake and symptom onset. Accordingly, in a large sample of self-reported lactose-intolerant patients, we recorded the symptom pattern considered as lactose related by the patients, and we compared the presence of these symptoms in patients with and without lactose malabsorption. Moreover, in a subgroup of randomly selected patients, first an in-depth psychological evaluation through validated questionnaires was performed, and subsequently, in a double-blinded protocol, the occurrence of symptoms after 20 g lactose solution and 1 g glucose solution was tested. 

## 2. Materials and Methods

A group of 268 consecutive patients (204 females, mean age 36 ± 10 years) undergoing lactose breath test was enrolled. Ninety-seven patients fulfilled Rome IV criteria for functional dyspepsia, 50 for irritable bowel syndrome, 27 for functional diarrhoea, and 27 for functional constipation. All the patients fulfilled diagnostic criteria for functional bloating [10,11]. According to the Montreal Classification [12], 52 patients showed gastro-oesophageal reflux disease with oesophageal syndrome. Upper endoscopy was performed in 34 patients: six patients presented Los Angeles Class A esophagitis. This group of 268 patients was enrolled in the first part of the study, which consisted of the evaluation of symptoms considered as lactose related by self-reported lactose-intolerant patients. This large group of patients consisted of consecutive patients without organic conditions, referred by both general practitioners and gastroenterologists for the evaluation of symptoms that the patients attributed to the intake of lactose. 

In this group of 268 patients, a computer-generated randomization list (1:2) was applied, and a subgroup of 40 patients (35 females, mean age 39 ± 16 years) was selected. These patients participated in the second part of the study, aimed at the evaluation of the accuracy of symptoms occurring after oral lactose load to diagnose lactose intolerance.

A hydrogen breath tests on two different days after blinded administration of lactose or glucose was performed. The patients were informed of the inability of the low dose of glucose to induce symptoms and of the blinded administration. Moreover, the psychological characteristics of patients with and without lactose malabsorption was assessed through the administration of validated questionnaires.

In all patients, the presence of organic conditions was excluded by endoscopic or radiologic procedures, routine blood tests including thyroid function tests, coeliac disease associated serology, and abdominal ultrasound. Patients with Helicobacter pylori infection and allergic conditions were excluded. Patients treated with drugs known to interfere with intestinal function during the three months prior to the study, such as antibiotics, were also excluded. Once per week intake of prokinetics in dyspeptic patients, a gentle water enema when needed in constipated patients and, in patients with diarrhoea, loperamide once per week, if strictly necessary, were allowed. In patients with gastro-oesophageal reflux disease, the use of PPI was not allowed in the month before the enrolment. 

The protocol was approved by the local Ethical Committee (Prot. Dir. Scient. 20110002489), and all the subjects gave their written informed consent before participation.

### 2.1. Hydrogen Breath Tests

In order to avoid intestinal gas production, due to non-absorbable or slowly fermentable carbohydrate residuals in the colon, patients were asked to consume a dinner consisting of only rice, meat and olive oil [13] the evening before the test day. After a 12 h fasting period, breath testing started between 08.30 a.m. and 09.30 a.m., after thorough mouth washing with 40 mL of 1% chlorhexidine solution [14]. Smoking [15] and physical exercise [16] were not allowed for one hour prior to and throughout the test.

Sampling of alveolar air was performed by means of a commercial device (Gasampler Quintron, Milwaukee, WI, USA) that allows the first 500 mL of dead space air to be separated and discarded, while the remaining 700 mL of end-alveolar air was collected in a gas-tight bag. Subjects were instructed to avoid deep inspiration and not to hyperventilate before exhalation. The variability of H_2_ determinations using this collecting method was demonstrated to be about 25% [17]. A gas chromatograph dedicated to the detection of H_2_ and CH_4_ in air samples was used for breath sample analysis (Model DP12, Quintron Instrument, Milwaukee, WI, USA). The accuracy of the detector was ±2 ppm with a linear response range between 2 and 150 ppm of H_2_ and between 2 and 50 ppm of CH_4_. 

In the group of 268 subjects, involved in the first part of the study (Part A), the test solution of the H_2_/CH_4_ breath test consisted of 20 g of lactose dissolved in 400 mL of water. Breath samples were taken at fasting and every 30 min for a seven-hour period. The test was considered as indicative of the presence of lactose malabsorption when the sum of the H_2_ concentration in breath samples obtained at the 5 ^a^, 6 ^a^ and 7 ^a^ h was ≥15 ppm [18]. 

In the group of 40 subjects, involved in the second part of the study (Part B), the two H_2_/CH_4_ breath test solutions consisted of 20 g of lactose dissolved in 400 mL of water or 1g of glucose dissolved in 400 mL of water. In negative patients, on a separate day, the H_2_/CH_4_ breath test after lactulose administration was also performed to evaluate the intestinal gas production capacity. Twenty grams of lactulose in 400 mL of water was orally administered, and breath samples were collected at fasting and every 15 min for an 8 h period. Subjects were considered H_2_ producers if H_2_ breath excretion exceeding 20 ppm was detectable during the test [19] and CH_4_ producers if CH_4_ breath excretion of more than 5 ppm was detectable [20].

### 2.2. Recognition of Lactose Solution

In a preliminary evaluation, in order to establish the blinding for the Part B study, we tested whether the two solutions could be identifiable by a group of twenty healthy volunteers, members of the medical and paramedical staff (12 females, mean age 31 ± 5 years, range 27–38). On two different days, in a random order, the healthy volunteers were asked to drink the two solutions and clearly state which one was the lactose and which one the glucose solution. The proportion of subjects who correctly identified the lactose solution was not different from the proportion of subjects who did not (Figure 1). 

### 2.3. Symptom Evaluation

In the group of 268 patients involved in the first part of the study (Part A), the presence and severity of symptoms that enrolled patients considered induced by lactose intake was investigated. We used a visual analogue scale, ranging from 0 (no distress) to 10 (highest distress) [21]. The symptom was considered mild when the score was 3 or less, moderate when higher than 3 and lower than 6, severe when 6 or higher. The investigated dyspeptic symptoms were epigastric pain, epigastric burning, early satiety, fullness, nausea, vomiting, belching. The investigated reflux-related symptoms were heartburn and regurgitation. Among lactose intolerance symptoms, abdominal pain, abdominal distention, bloating, and flatulence were investigated together with headache, an extra-abdominal symptom. Bowel habit was also investigated: a subject was considered as constipated if >25% of the evacuations were characterized by straining and stools were rated as type 1 or 2 of the Bristol stool form scale; a subject was considered as diarrhoeic if >25% of the evacuations were characterized by type 6 or 7 of the Bristol stool form scale [11].

In the group of 40 patients involved in the second part of the study (Part B), the presence and severity of symptoms during the two hydrogen breath tests were monitored by visual analogue scale administration every 15 min for the 7 h test duration, as in the first part of the study (Part A). We investigated some extraintestinal symptoms, such as tiredness, foggy mind, feeling of empty head, sleepiness, besides headache.

### 2.4. Questionnaires

Each subject of the Part B study was evaluated by means of the following questionnaires. The Multidimensional Iowa Suggestibility Scale (MISS) [22] is a 95-item self-reported questionnaire investigating suggestibility rated from 1 (“Not at all or slightly”) to 5 (“A lot”). The MISS includes five suggestibility subscales: Consumer Suggestibility (CS), Persuadability (PER), Physiological Suggestibility (PS), Physiological Reactivity (PHR), Peer Conformity (PC) and two companion scales (Psychosomatic Control, Stubborn Opinionatedness). Higher scores indicate higher suggestibility.

The State-Trait Anxiety Inventory-Y form (STAI-Y) [23] is composed of two separate scales of 20 items each: the STAI-Y form 1 measures present levels of anxiety (state anxiety), while the STAI-Y form 2 evaluates general anxiety traits (trait anxiety). Participants are asked to indicate how they “generally feel” on a four-point Likert scale ranging from 1 to 4. Higher scores indicate higher anxiety levels.

The Big Five Inventory [24] is a 44-item self-administered scales used to measure the following personality domains: (1) neuroticism, which refers to the propensity to experience anxiety, hostility, impulsivity, depression and psychological distress in response to different stressors; (2) extraversion, which refers to people who are sociable, enthusiastic and energetic; (3) openness to experience, which refers to the tendency of being curious, innovative, imaginative and unconventional; (4) agreeableness, which denotes people who are straightforward, forgiving, modest and sympathetic; (5) conscientiousness, which refers to the tendency of being competent, organized, scrupulous and dutiful. Items are rated on 5-point Likert-type scale where 1 means strongly disagree and 5 strongly agree. Higher scores on a scale imply stronger representation of the feature.

### 2.5. Statistics

Descriptive statistics were calculated for all variables. Normality of the assessed measures was tested by Kolmogorov–Smirnoff test. As all the included variables were not normally distributed, non-parametric tests were applied to compare the two groups. Correlation between variables of interest was evaluated by means of Spearman’s rho. Multiple linear regression analyses were performed to identify independent predictors of specific gastrointestinal symptoms or development of nocebo in patients. Presence of nocebo was defined as the development of symptoms during glucose administration. 

Symptoms were considered significant if they exceeded 30% increase from the VAS evaluation at baseline. All calculations were performed with Stata 16.0 (StataCorp. 2019, College Station, TX, USA). A two-tailed *p* value < 0.05 was regarded as statistically significant.

## 3. Results

### 3.1. Part A

Lactose malabsorption, according to the positivity of lactose breath test, was present in 172 (64%) patients, while the test proved to be negative in 96 (36%) patients. In Table 1, the prevalence of functional dyspepsia, GERD, IBS, functional diarrhoea and functional constipation in patients with and without lactose malabsorption is shown. 

The prevalence of functional dyspepsia, IBS and functional constipation was significantly higher in the group of patients with negative breath test than in the group of lactose malabsorbers (*p* < 0.05 for all the comparisons). Figure 2 depicts the prevalence of moderate-to-severe symptoms considered by the group of 268 patients as caused by lactose intake during their day-to-day life. 

It is evident that abdominal symptoms, commonly attributed to lactose intolerance, are markedly more prevalent than symptoms related to functional dyspepsia or gastroesophageal reflux disease. However, these symptoms account for a prevalence ranging from 10% to 20% of cases. In Figure 3, the prevalence of moderate-to-severe symptoms according to the result of the lactose breath test is shown. 

In patients with a positive breath test (panel A), the prevalence of symptoms commonly related to lactose malabsorption significantly increased, and the prevalence of symptoms related to functional dyspepsia and gastroesophageal reflux disease decreased. On the contrary, in patients with a negative breath test (panel B), the prevalence of symptoms commonly related to lactose malabsorption significantly decreased, and the prevalence of symptoms related to functional dyspepsia and gastroesophageal reflux disease increased. The prevalence of bloating did not show modifications. 

In Table 2, the severity of dyspeptic symptoms and GERD-related symptoms in patients with and without lactose malabsorption is shown. 

In comparison with lactose malabsorbers, the severity of satiety, fullness and nausea was significantly higher in the group of patients with negative breath test (*p* < 0.005 for all the comparisons). Similarly, in comparison with lactose malabsorbers, the severity of heartburn and regurgitation was significantly higher in patients without lactose malabsorption (*p* < 0.001 for both comparisons).

As far as headache is concerned, in comparison with lactose malabsorbers, the prevalence of this symptom was significantly higher in patients without lactose malabsorption (24% vs. 12%, *p* = 0.016). 

In Table 3, the severity of abdominal pain, abdominal distention, bloating, and flatulence is shown. 

In comparison with the subgroup of patients with negative breath test after lactose administration, patients with a positive test showed high severity of abdominal pain, abdominal distention and flatulence. 

### 3.2. Part B

Among the 40 tested patients, 27 proved to be lactose malabsorbers, and 13 showed a normal lactose absorption. Figure 4 shows the relationship between lactose or glucose intake, administered in a blinded fashion, and symptom reporting by patients with and without lactose malabsorption. 

In the subgroup of 27 patients with lactose malabsorption, ten patients reported symptoms only after lactose, configuring a diagnosis of lactose intolerance. On the contrary, symptoms occurred in four patients after glucose and in eight patients after both lactose and glucose. In these 12 patients, it was not possible to ascertain the relationship between carbohydrate intake and symptom occurrence. Five patients did not report symptoms after lactose or glucose administration, configuring a subgroup of lactose-tolerant patients with malabsorption.

In the subgroup with normal lactose breath test, four patients did not report symptoms, and we must consider them as lactose-tolerant subjects. On the contrary, three patients reported symptoms only after lactose, one patient only after glucose and five patients after both lactose and glucose. Again, in these patients, it was not possible to define the relationship between lactose intake and symptom occurrence. Therefore, in 21 out of 40 patients (52.5%), hydrogen breath test was inaccurate to select intolerant patients, even if a double blind protocol to avoid nocebo effect was adopted.

No difference in gender between patients showing and not showing nocebo effect was detected (% females 51.5 vs. 48.5, Chi-square = 0.23, *p* = 1.0). Moreover, considering questionnaires, we did not observe any significant differences between patients showing and not showing nocebo effect (Table 4). 

Additionally, it was not possible to create a significant logistic regression model of nocebo respondents. Finally, patients with positive lactose breath test did not show a comparison with patients with a negative lactose breath test.

## 4. Discussion

The diagnosis of food intolerance is complex for the clinician. Patients with symptom onset after meal ingestion are common: they tend to blame several foods as responsible for symptoms and consider themselves as “intolerant to all foods”. These patients frequently misinterpret the origin of their symptoms and self-diagnose a food intolerance, generally not present [25], caused instead by a functional gastrointestinal disorder. In our experience, the extremely high number of foods regarded as culprit by the single patient strongly suggests sensorimotor abnormalities of the gastrointestinal tract as responsible for the origin of post-prandial symptoms rather than food intolerance [25,26,27]. Moreover, important co-factors facilitating an incorrect interpretation by patients about the relationship between food intake and symptom onset are the overlap of intolerance symptoms with functional gastrointestinal disorders and the absence of validated diagnostic tests to diagnose food intolerances.

Nevertheless, even when the presence of accurate tests may be of help, the diagnosis may be a minefield: lactose malabsorption may be diagnosed accurately with hydrogen breath test [4,8]. On the contrary, the diagnosis of lactose intolerance is difficult, due to the inaccuracy of symptom monitoring after unblinded oral lactose load [4]. Since the prevalence of lactose malabsorption is high in the Italian population [28,29], the clinical value of a positive hydrogen breath test is low, due to the unavailability of an accurate test for lactose intolerance, but the negative predictive value becomes high. Accordingly, in clinical practice, the hydrogen breath test after lactose oral load allows for an accurate exclusion of lactose intolerance in the pathophysiology of patient symptoms. 

Furthermore, it is extremely important to define which symptoms depend on lactose intake and to clarify to patients the correct relationship between the effect of malabsorbed lactose and the pathophysiology of evoked symptoms. Our findings showed that, in comparison with patients with lactose malabsorption, patients without lactose malabsorption, and obviously without lactose intolerance, more frequently attributed to lactose ingestion the onset of symptoms deriving from functional gastrointestinal disorders. In particular, presence and severity of both dyspeptic and GERD-related symptoms were significantly higher in patients with negative lactose breath test in comparison with patients with a positive breath test. On the contrary, as expected, abdominal pain, bloating and flatulence proved to be more prevalent in patients with a positive breath test. Besides the well-known inaccuracy of self-reported lactose intolerance [6], these results point out that patients erroneously attribute to lactose intake symptoms of other functional gastrointestinal disorders [26]. It is conceivable that this is caused by the lack of information about the effects of malabsorbed lactose on the gastrointestinal tract. Many patients ignore the intraluminal consequence of malabsorbed lactose representing the main mechanism for lactose intolerance: lactose fermentation at colonic level, the consequent production of gases, water, short chain fatty acids and, in turn, the onset of diarrhoea, flatulence, abdominal pain, and acidification of stools. Thus, patients attribute to lactose malabsorption every symptom occurring after dairy product intake, interpreting as lactose dependent also symptoms related to lipidic content of dairy products, such as, for instance, postprandial fullness or heartburn. The potential for lactose malabsorption to be responsible for extra-intestinal symptoms is still a matter of debate: specific mechanisms able to induce these symptoms are still unknown.

This error has two main consequences: first, a diagnostic delay of the actual condition responsible for patient symptoms resulting in an impaired quality of life. Additionally, since lactase deficiency may also be secondary to several enteropathies, the diagnostic delay could cause severe nutritional deficits. Second, patients may inappropriately withdraw lactose from diet and expose themselves to the consequences of a negative calcium balance [30,31]. Therefore, fostering correct information about symptomatology of lactose malabsorption through clear media and social campaign represents an important goal for the next decades. 

The second part of our study showed that the blinded administration of lactose or glucose allows for a correct definition of the relationship between symptom onset and lactose intake only in 47.5% of the sample, while a nocebo effect occurred in the other subgroup of patients. Our working hypothesis was that the blinded administration of lactose could avoid the interfering role of nocebo effect and that lactose intolerance could be correctly unmasked. The results disprove this hypothesis. Moreover, such a complex protocol could be of difficult application in routinely clinical practice. The importance of lactose intolerance come to light if we consider the high consumption of milk and dairy products in Europe. Data concerning dairy product consumption in 2020 [32,33] show that the European Union is in sixth place in the world for milk consumption, with 65.13 kg pro capite, +0.36% in comparison with 2019. As far as cheese consumption is concerned, the EU is in first place with 18.39 kg pro capite, +0.46% in comparison with 2019. Therefore, on the basis of these data, the development of an accurate test to diagnose lactose intolerance represents an unmet need.

We analysed the psychological profile of patients undergoing lactose breath test to verify if some psychological characteristics may be of use in the selection of patients exposed to nocebo effect. None of the studied characteristics (personality traits, suggestibility, and anxiety) were clearly related to the presence of nocebo effect.

This is in line with a recent paper, showing no effect of anxiety or personality traits on the onset of nocebo effect [34,35]. Of note, we did not observe an effect of suggestibility on the development of nocebo response as in a previous study [36]. However, our study was not specifically designed to elicit strong nocebo response related to somatic pain as in the aforementioned study. Our study was conducted in a clinical setting and not in a laboratory one. Our measure of suggestibility was a self-reported questionnaire and therefore was subjected to several biases: for instance, people self-diagnosed with lactose intolerance may have already been labelled as suggestible and therefore would have answered the questionnaire according to a desirability bias.

Two previous studies analysed the relationship between psychological factors and the perception and reporting of symptoms after lactose intake [37,38]. Suarez et al. demonstrated that in comparison with patients without lactose intolerance, in self-reported lactose-intolerant patients, the ingestion of two cups of milk was not associated with a significantly higher score of symptoms. Unfortunately, in this study, the evaluation of psychologic factors was inconclusive, due to a high rate of dissimulation. In another paper [38], in comparison with patients without intolerance, lactose-intolerant patients showed a higher severity of somatization. In this latter paper, lactose breath test was performed with a different protocol, consisting of 3 h of breath sampling after the administration of a low dose of lactose, 15 g. It was shown that the accuracy of a hydrogen breath test increases by increasing its duration [18,19], and an imperfect separation between lactose malabsorbers and absorbers was expected in the paper by Tomba et al. Moreover, the administration of a low dose of lactose reduces the number of positive patients but does not reduce the impact of nocebo effect, causing an imperfect categorization of patients in lactose tolerants and lactose intolerants and, in turn, an imperfect evaluation of the effect of psychological factors. 

Our study has several limitations that should caution against over-interpretation: Firstly, the sample size of the cohort involved in the second part of the study is quite small and is mostly composed of women; as a consequence, some limits for the generalization of our results to other populations can be raised. Secondly, the use of self-report scales without a formal psychiatric assessment could have underestimated the presence of psychological conditions. Finally, we did not assess visceral hypersensitivity, a condition known to modify the sensitive response to intestinal stimuli which could play a key role in the clinical presentation of lactose intolerance [39]. 

## 5. Conclusions

In conclusion, our study showed that, frequently, patients are not sufficiently informed on the correct symptom pattern of lactose intolerance and erroneously attribute to lactose ingestion the occurrence of dyspeptic- and reflux-related symptoms. As the presence of nocebo effect could be a potential confounding factor in the diagnosis of lactose intolerance, the development of new strategies to avoid nocebo response should be of primary importance both in research and in clinical practice.

## Figures and Tables

**Figure 1 ijerph-19-10234-f001:**
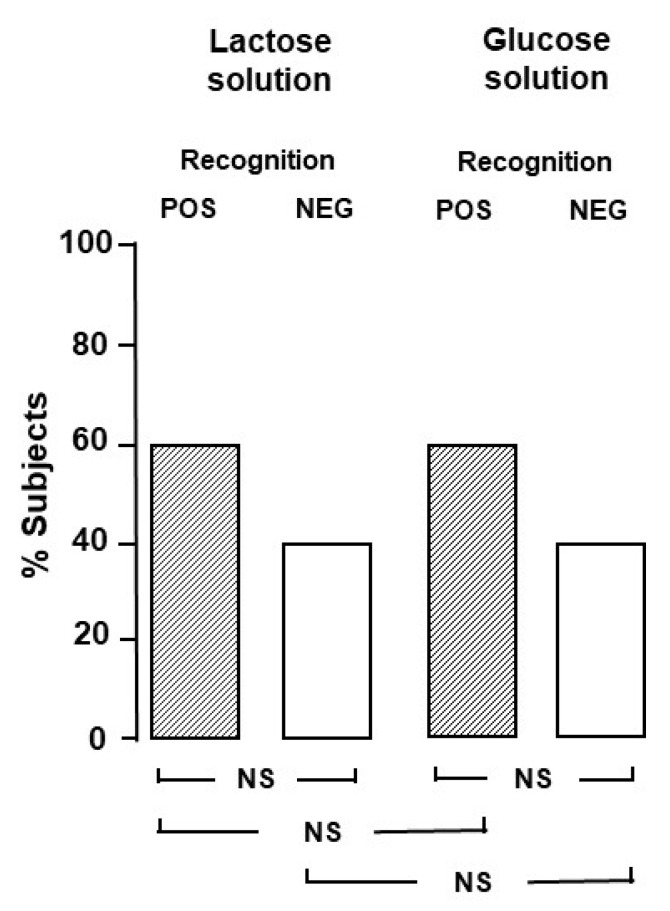
Recognition of lactose and glucose solutions used for blinded breath test in healthy volunteers.

**Figure 2 ijerph-19-10234-f002:**
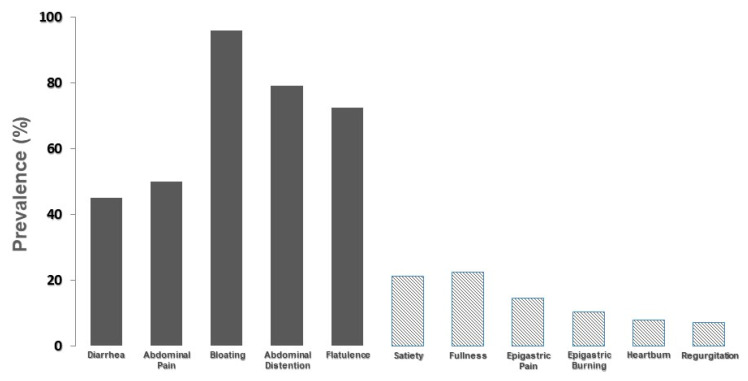
Prevalence of moderate-to-severe symptoms attributed to the ingestion of lactose by the group of 268 patients. Grey bars indicate intolerance-related symptoms; bars with diagonal lines indicate dyspeptic- and reflux-related symptoms.

**Figure 3 ijerph-19-10234-f003:**
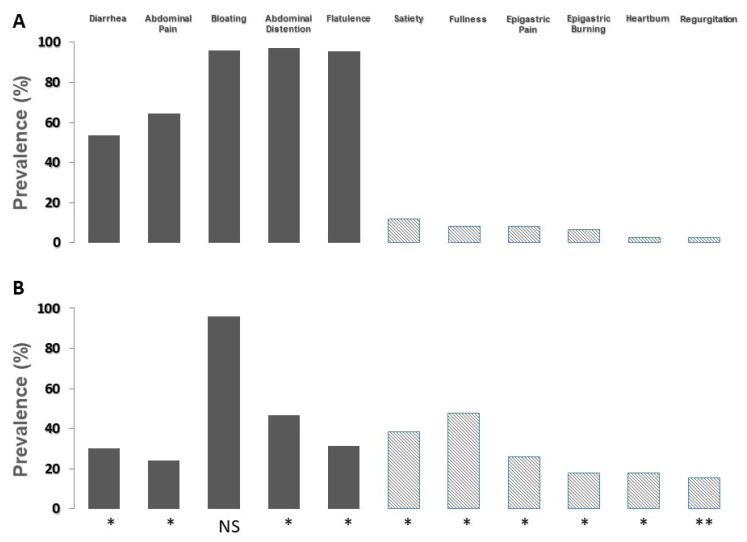
Prevalence of moderate-to-severe symptoms attributed to the ingestion of lactose by the group of 268 patients, according to lactose breath test results. (**A**) Patients with positive lactose breath test. (**B**) Patients with negative lactose breath test. * = *p* < 0.0001 and ** = *p* < 0.007 for the comparison of the severity of each symptom between patients with positive and negative lactose breath test. Grey bars indicate intolerance-related symptoms; bars with diagonal lines indicate dyspeptic- and reflux-related symptom.

**Figure 4 ijerph-19-10234-f004:**
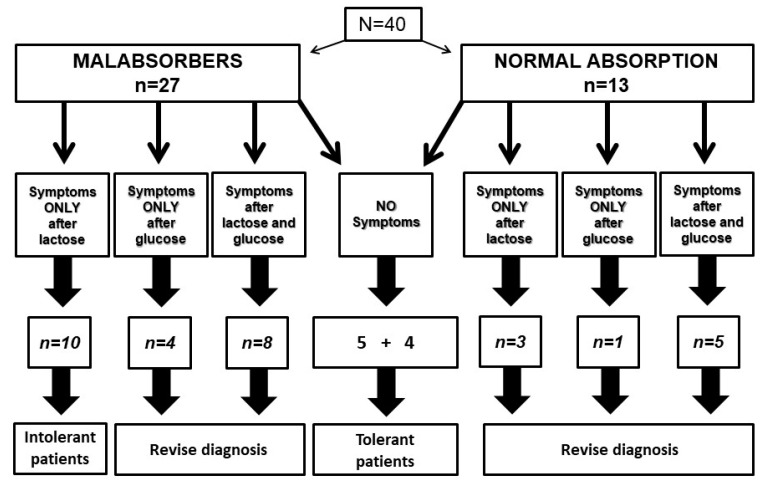
Results of blinded lactose and glucose breath tests in 40 patients self-reporting lactose intolerance.

**Table 1 ijerph-19-10234-t001:** Prevalence of functional dyspepsia, GERD, IBS, functional diarrhoea and functional constipation in patients with and without lactose malabsorption.

	Lactose Breath Test Positive n (%)	Lactose Breath Test Negative n (%)	*p*
Functional dyspepsia	48 (28%)	49 (49%)	0.0002
GERD	34 (20%)	18 (19%)	NS
IBS	24 (14%)	26 (27%)	0.013
Functional diarrhoea	14 (8%)	13 (13%)	NS
Functional constipation	18 (10%)	19 (20%)	0.042

**Table 2 ijerph-19-10234-t002:** Severity of dyspeptic and GERD-related symptoms in patients with and without lactose malabsorption.

	Lactose Breath Test Negative	Lactose Breath Test Positive	*p*
*Dyspeptic Symptoms*			
Epigastric pain	1.2 ± 1.7	1.0 ± 1.2	ns
Epigastric burning	0.8 ± 1.2	0.6 ± 0.9	ns
Satiety	5.2 ± 2.2	4.3 ± 2.2	0.001
Fullness	5.8 ± 1.6	3.3 ± 1.8	0.0001
Nausea	2.4 ± 1.7	1.6 ± 1.5	0.006
Vomiting	0.1 ± 0.1	0.1 ± 0.1	ns
Belching	0.1 ± 0.1	0.1 ± 0.1	ns
*GERD-related symptoms*			
Heartburn	6.1 ± 1.6	2.7 ± 1.0	0.0001
Regurgitation	5.6 ± 1.8	2.8 ± 1.2	0.0001

**Table 3 ijerph-19-10234-t003:** Severity of common lactose intolerance symptoms in patients with and without lactose malabsorption.

	Lactose Breath Test Negative	Lactose Breath Test Positive	*p*
Abdominal pain	1.3 ± 2.7	3.9 ± 2.5	0.001
Abdominal distention	3.2 ± 1.6	6.5 ± 1.8	0.001
Bloating	6.5 ± 2.1	6.3 ± 2.1	ns
Flatulence	2.9 ± 1.5	6.2 ± 1.9	0.001

**Table 4 ijerph-19-10234-t004:** Psychological questionnaire scores between patients with and without nocebo effect.

	Nocebo Positive *Mean ± SD*	Nocebo Negative *Mean ± SD*	*p*
*MISS*			
Consumer Suggestibility	19.16 ± 5.43	20.18 ± 7.35	0.87
Persuadability	32.58 ± 6.63	34.53 ± 7.03	0.51
Physiological Suggestibility	16.32 ± 4.22	18.88 ± 5.13	0.07
Physiological Reactivity	38.53 ± 9.11	41.47 ± 6.78	0.30
Peer Conformity	34.89 ± 5.21	34.71 ± 5.96	0.71
Psychosomatic Control	29.47 ± 6.80	28.35 ± 6.49	0.59
Stubborn Opinionatedness	41.53 ± 8.93	42.35 ± 10.96	0.90
Total MISS	141.47 ± 23.41	149.76 ± 23.70	0.27
*STAI-Y*			
State Anxiety	46.47 ± 3.08	46.59 ± 2.21	0.73
Trait Anxiety	50.94 ± 2.98	49.71 ± 2.80	0.13
*Big Five Inventory*			
Neuroticism	25.79 ± 3.95	26.32 ± 4.19	0.66
Extraversion	24.32 ± 4.47	23.11 ± 5.27	0.51
Openness to experience	33.84 ± 5.14	32.32 ± 8.76	0.79
Agreeableness	30.00 ± 3.43	28.79 ± 4.08	0.34
Conscientiousness	32.32 ± 5.61	31.84 ± 5.14	0.62

## Data Availability

The data presented in this study are available on request from the corresponding author.

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
