# Peer review of "The Perception of Lactose-Related Symptoms of Patients with Lactose Malabsorption"

_ijerph, 2022, doi:10.3390/ijerph191610234_

Round 1

Reviewer 1 Report

Thank you for opportunity to review the paper:

“The Perception of Lactose-related Symptoms of Patients with Lactose Malabsorption”

The study was well designed and the methods are described properly.

The paper need to be revied concerning minor mistakes/ English language style.

For instance:

In Table 1: 19 (20) should be replaced by 19 (20%).

In discussion: …it is very common to  visit outpatients reporting onset of symptoms in the postprandial period… style should be corrected

I still recommend the paper for publication after minor revisions

Author Response

We thank reviewer 1 for his/her comments.

We have not made the requested modifications of English language: a revision by a mother language translator was performed. 

Reviewer 2 Report

The reviewed manuscript ijerph-1811725 entitled “The Perception of Lactose-related Symptoms of Patients with Lactose Malabsorption” by Di Stefano and co-authors describes the research aimed to analysed which symptoms patients consider lactose-related and evaluated if psychological profiles may affect the interpretation of the relationship between lactose and symptom. 

The manuscript presents very interesting results of a very well-planned and conducted clinical trial. In conclusion, the authors indicated that patients with self-reported lactose intolerance are frequently unaware of the clinical presentation and symptom patterns of this condition thus, very frequently they incorrectly attribute the occurrence of dyspeptic and reflux-related symptoms to lactose ingestion.

These results are of great importance and this manuscript will open a wide discussion between specialists in gastroenterology, allergy and psychology but should also reach the hands of primary care physicians, both paediatrics and internists.

In my opinion, this manuscript has great scientific value, also the subject is very current. The results presented in this manuscript could give important clues which may potentially significantly shorten the correct lactose malabsorption or lactose intolerance diagnosis and thus be helpful in faster the treatment of the patient but on the other hand, it could help to reduce the cost of healthcare.

The authors rightly point out that proper diagnosis requires precise guidance and validated diagnostic tests. I suggest accepting it for publication in its present form although I could suggest two minor changes:

- instead of cubic centimetres better to simply use ml of water as 1 cc is almost equal to 1 ml - please uniform the units in lines 127, 129 and 131;

- it could be easier for a reader to distinguish the symptoms related to lactose malabsorption from those related to functional dyspepsia and gastroesophageal reflux symptoms if the bars in Fig 3. would be diversified by colour or texture

Author Response

We thank reviewer 2 for his/her comments.

We have not made the requested modifications of English language: a revision by a mother language translator was performed. 

We have now used ml instead of cc and we have now indicated dyspeptic and refux related symptoms in the figures with a differente bar.